# QUANTBENCH: BENCHMARKING AI METHODS FOR QUANTITATIVE INVESTMENT

## ABSTRACT

The field of artificial intelligence (AI) in quantitative investment has seen significant advancements, yet it lacks a standardized benchmark aligned with industry practices. This gap hinders research progress and limits the practical application of academic innovations. We present QuantBench, an industrial-grade benchmark platform designed to address this critical need. QuantBench offers three key strengths: (1) standardization that aligns with quantitative investment industry practices, (2) flexibility to integrate various AI algorithms, and (3) full-pipeline coverage of the entire quantitative investment process. Our empirical studies using QuantBench reveal some critical research directions, including the need for continual learning to address distribution shifts, improved methods for modeling relational financial data, and more robust approaches to mitigate overfitting in low signal-to-noise environments. By providing a common ground for evaluation and fostering collaboration between researchers and practitioners, QuantBench aims to accelerate progress in AI for quantitative investment, similar to the impact of benchmark platforms in computer vision and natural language processing.

## 1 INTRODUCTION

Although artificial intelligence (AI) for quantitative investment has been extensively studied by research communities and many new trading algorithms have been developed in recent years, there is a lack of a standardized benchmark that aligns with the testing standards used in the quantitative investment industry and by numerous trading firms. The absence of a standard benchmark dataset and test pipeline, coupled with the diverse standards used in existing papers to evaluate algorithm performance, may hinder research progress and limit the practical application of these advancements in the industry. Meanwhile, the establishment of standardized benchmarks, as demonstrated by some representative examples in computer vision (Deng et al., 2009) and natural language processing (Wang et al., 2019a), has proven instrumental in advancing research in their respective fields.

To address this need, we propose QuantBench, an industrial-grade benchmark platform that offers universality and comprehensive pipeline coverage. QuantBench offers (1) **Standardization**: QuantBench adheres to research standards that are consistent with industrial practices in quantitative investment. (2) **Flexibility**: QuantBench is designed to support the scalable integration of various AI algorithms into the system. (3) **Full-pipeline Coverage**: QuantBench encompasses the entire pipeline of general quantitative investment strategies, offering broad support for standardized datasets, model implementations, and evaluations.

Specifically, QuantBench employs a layered approach to the quant research pipeline, as shown in Figure 1, integrating diverse tasks and learning paradigms into a single platform. By incorporating a broad array of models, QuantBench bridges the gaps created by the segmented nature of the field, enhances reproducibility through open-source implementations, and facilitates the integration of advancements across disciplines. In addition, QuantBench also emphasizes the importance of a unified dataset, constructing datasets with both breadth and depth while maintaining consistency in data format across varied data types. Moreover, QuantBench offers a detailed market simulation framework that can vary in realism depending on the specific trading scenario. This robust evaluation framework accommodates quant-specific metrics that are both task-specific (e.g., signal and portfolio performance) and task-agnostic (e.g., alpha correlation and decay). By aligning evaluation metrics with tasks, QuantBench ensures that the chosen metrics capture the intricacies of these

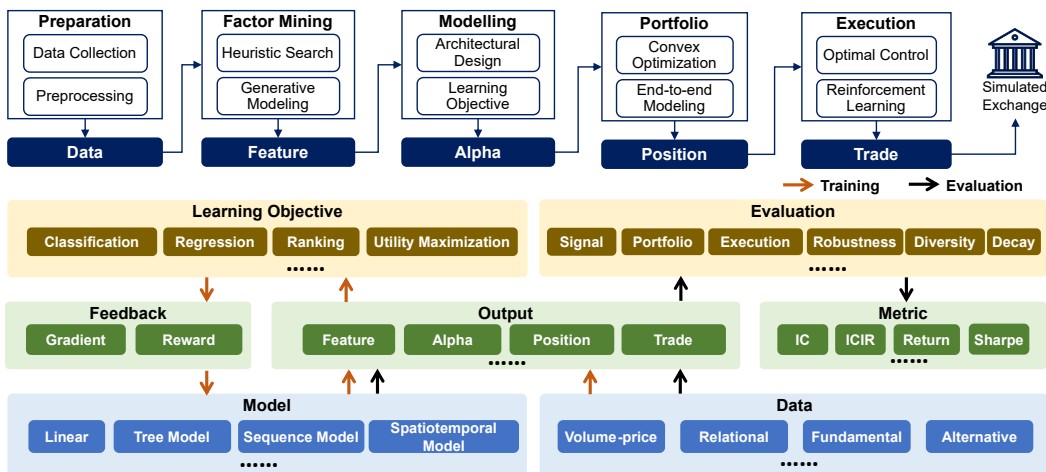

Figure 1: Overview of QuantBench. **Upper**: Quant research pipeline covered in QuantBench. **Lower**: The layered design of QuantBench.

tasks, thereby enhancing the relevance of the findings. Additionally, it pays careful attention to task-agnostic metrics to address the unique challenges presented by financial data, including low signal-to-noise ratios and non-stationarity.

The contribution of QuantBench can be summarized as follows:

1. QuantBench enhances research efficiency by alleviating researchers from time-consuming pre-processing tasks, enabling them to focus on critical algorithmic innovations.

2. QuantBench serves as a unified platform. Industry practitioners can leverage this platform to rapidly implement state-of-the-art algorithms in their investment strategies. Conversely, QuantBench amplifies the impact of academic research by facilitating its practical application.

3. Through the standardization provided by QuantBench, communication between academia and industry is improved, thereby accelerating progress in the field of AI for quantitative investment.

4. Our empirical studies using QuantBench have identified several compelling research directions with significant potential value: a) The distribution shift problem in quant data leads to rapid model degradation, highlighting the need for continual learning approaches for efficient model updates. b) Incorporation of graph structures does not consistently yield performance improvements, indicating a need for more sophisticated methods to represent and model relational data in financial contexts. c) While deep neural networks excel at fitting training objectives, this proficiency does not necessarily translate to superior returns over tree models, suggesting a potential misalignment between typical training goals and real-world performance metrics. d) Model ensembles show promise in mitigating overfitting issues caused by the low signal-to-noise ratio inherent in quantitative data. However, this finding also underscores the need for more robust modeling approaches, such as training models with inherent diversity or employing causal learning techniques.

## 2 THE QUANT PIPELINE

QuantBench facilitates the evaluation of the entire research pipeline as illustrated in Figure 1. The platform integrates features for data preparation and simulated trading environments, supporting the four key phases of research:

- **Factor Mining:** This process involves identifying predictive financial features (Tulchinsky, 2019). Within QuantBench, we enable formula-based factor mining, where each factor is defined as a symbolic equation combining raw data elements and operational functions. Methods such as evolutionary algorithms (Zhang et al., 2020; Cui et al., 2021) are utilized to devise these expressions. Addi-

Table 1: Comparison of different tasks covered in QuantBench

| Task | Data | Output | Objective | Feedback | Eval |
|------|------|--------|-----------|----------|------|
| Factor Mining | Data | Features | Regression | Reward | Signal |
| Modelling | Features | Prediction | Regression/Classification/Ranking | Gradient | Signal |
| End-to-end Modeling | Features | Position | Utility Maximization | Gradient | Portfolio |
| Portfolio Optimization | Prediction | Position | Utility Maximization | Reward | Portfolio |
| Order Execution | Position | Trade | Utility Maximization | Reward | Execution |

tionally, reinforcement learning techniques (Yu et al., 2023) have been employed to autonomously discover high-performing factors through policy gradient methods.

- **Modeling:** This phase involves constructing models to forecast market movements (classification) (Koa et al., 2024), predict asset returns (regression), or identify the most or least valuable assets (ranking) (Feng et al., 2019). A variety of machine learning and deep learning approaches are applicable in this stage, where the inputs are the features identified in the factor mining phase.

- **Portfolio Optimization:** This stage seeks to determine the optimal asset allocation to maximize an investor's utility, which is defined based on their risk profile. Simple strategies may involve charaterstic-sorted portfolios (Cattaneo et al., 2020), where trading decisions are made according to predicted values such as asset returns. More sophisticated approaches, such as mean-variance optimization (Markowitz, 1952; 1959), aim to balance risk against return. Recent works also explore the use of deep learning models to directly generate portfolio allocations in an end-to-end approach, training the model to maximize utility.

- **Order Execution:** The goal here is to execute buy or sell orders at optimal prices while minimizing market impact. Placing large orders at once can adversely affect the asset's price, thus increasing trading costs. Traditional strategies employ optimal control techniques (Bertsimas & Lo, 1998; Almgren & Chriss, 2000) to derive an execution strategy, while recent advancements have incorporated reinforcement learning (Nevmyvaka et al., 2006; Fang et al., 2021; 2023) to optimize this process.

## 2.1 DESIGN OF QUANTBENCH

QuantBench adopts a structured, layered approach, integrating all research phases into a comprehensive framework, as depicted in Figure 1. At the foundation, the bottom layer consists of a diverse array of models and datasets that underpin the entire quant research pipeline. This layer ensures that a broad spectrum of financial data types and modeling techniques are accessible for rigorous analysis. The middle layer of QuantBench processes outputs from the models, incorporates feedback mechanisms, and applies evaluation metrics. This layer effectively translates complex data and model interactions into quantifiable results that facilitate direct comparisons and iterative improvements in model performance. At the top, the framework outlines the learning objectives and the criteria for comprehensive evaluation. This layer aims to align the research outcomes with specific investment goals, such as maximizing utility or optimizing asset allocation, ensuring that the training and evaluation stages are directly relevant to real-world financial strategies.

Within this structured design, the training and evaluation processes follow distinct yet interconnected pathways. For training, the flow is: model + data → output → learning objective → feedback → model. For evaluation, the sequence is: model + data → output → evaluation → metric. These pathways ensure that both training and evaluation are systematic and aligned with the overarching objectives of the research. Table 1 offers a detailed comparison of each task according to this hierarchical structure.

## 3 DATA

Data serves as the fundamental source of information for quantitative predictions and decision-making in financial markets. The development of robust datasets in quantitative finance follows two primary directions: increasing breadth and enhancing depth. QuantBench addresses both aspects comprehensively. Increasing breadth involves incorporating diverse information sources, while enhancing depth focuses on improving data granularity. For instance, in low-frequency scenarios such

Figure 2: Data processing pipeline of QuantBench. Blocks with green background are already supported in QuantBench, and blocks with blue background are planned to be supported in the future.

as monthly portfolio rebalancing, the integration of alternative data sources (e.g., satellite imagery or credit card transaction data) can provide a more comprehensive context for each data point. Conversely, enhancing depth, such as transitioning from daily to minute-level stock price data, allows for the detection of subtle patterns and trends, thereby potentially improving trading outcomes through the preservation of more detailed information.

## 3.1 INCREASING DATA BREADTH

The expansion of information sources is crucial for developing a comprehensive view of the market. By integrating diverse data types, it becomes possible to uncover latent relationships and enhance predictive accuracy. QuantBench incorporates the following information sources:

- **Market data**: This includes price and volume data for stocks, options, and other financial instruments. These data form the basis for many technical analysis strategies and can reveal trends in market sentiment. In QuantBench we cover both aggregated bars (e.g. OHLCV) at regular time intervals, and tick-level trade/quote data at irregular timesteps. Our starting point of market data ranges from 2003 to 2006 and the current cutoff is May 2024.

- **Fundamental**: Fundamental data includes financial statements, earnings reports, and other company-specific information. These data provide insight into a company's underlying value and growth prospects. In QuantBench we collect fundamental data from the income statements, balance sheets, and statements of cash flows from publicly listed companies and construct 21 built-in features out of these data.

- **Relational**: Relational data captures the interactions between different entities, such as supply chain relationships or corporate ownership structures. We collect relational data from Wikidata by performing entity alignments based on company names and ticker symbols. We also construct fully-connect graph/hypergraph from industrial categorizations following the GICS categorization. Notably, to avoid future information leakage, we took the temporal information of relations from Wikidata into account and supports graph snapshots at any given timestamp. The statistics of relational data is shown in Table 9.

- **News**: News data, including articles, social media posts, and press releases, can provide real-time information on market-moving events and shifts in sentiment. In QuantBench we collect both the original news contents and processed numerical features such as news count and normalized sentiment scores.

Furthermore, QuantBench incorporates a diverse range of markets, universes, and feature sets to support more granular analysis:

- **Markets and Universe**: The dataset spans a range of markets, from those heavily influenced by automated trading to emerging markets, each exhibiting distinct trading patterns and dynamics. In markets dominated by automated trading, price adjustments tend to be swift and the trading patterns intricate, reflecting the rapid decision-making processes. Conversely, emerging markets often display slower price dynamics influenced significantly by individual investors. This diversity allows QuantBench to evaluate model performance across various market behaviors. Additionally, stocks are categorized into large-cap, mid-cap, and small-cap segments, each presenting

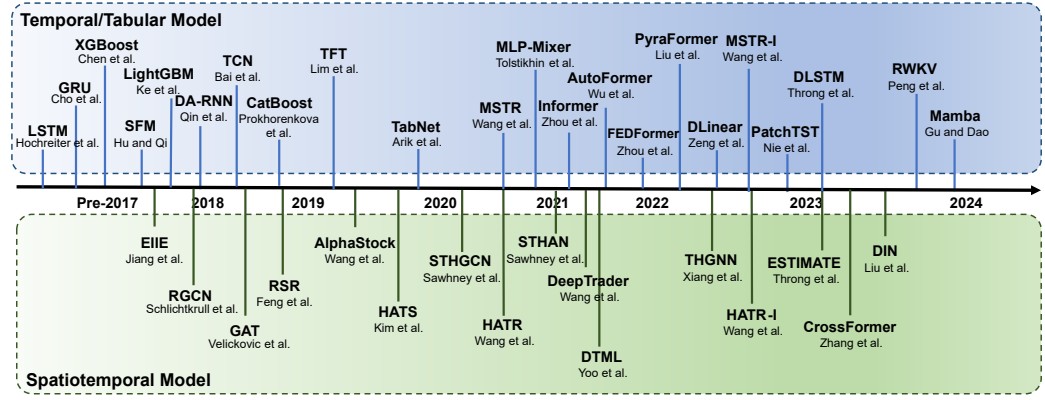

Figure 3: A non-exhaustive illustration of models covered in QuantBench and their evolution

unique characteristics. For example, large-cap stocks generally exhibit stability, contrasting with the higher volatility and potential for growth in mid-cap and small-cap stocks. Such categorization facilitates a detailed analysis of how market capitalization affects stock features and the predictive accuracy of models, thereby highlighting the differences in risk and returns across segments.

• **Feature Sets**: Leveraging QuantBench's capabilities in factor mining, we have integrated several widely-used feature sets at the daily level. These include Alpha158 (Yang et al., 2020), which offers a range of technical indicators, Alpha101 (Kakushadze, 2016), featuring short-term volume-price characteristics, and Alpha191 (Li & Liu, 2017), tailored specifically for the Chinese stock market and also focusing on short-term volume-price patterns.

## 3.2 GOING DEEP

Enhancing data resolution provides a detailed view of market dynamics. On a broader scale, quarterly or monthly data can uncover long-term trends and cyclical patterns, which are invaluable for strategic asset allocation and portfolio planning. On a finer scale, tick-level data captures the intricacies of intraday price movements and the effects of market microstructure, which are crucial for developing high-frequency trading strategies. The selection of modeling techniques is also influenced by data frequency, with high-frequency data often necessitating more computationally efficient and scalable methods. In QuantBench, data frequencies range from quarterly (fundamental data derived from financial statements) to tick-level (detailing trades and quotes), enabling a wide array of quant tasks. Lower frequency data suits applications like factor investing and risk modeling, while higher frequency data is essential for analyses such as order book scrutiny and trade execution optimization. The capability to integrate data at varying frequencies also facilitates the development of innovative multi-scale strategies. A comprehensive description of these data levels is available in the supplementary materials.

## 4 MODELS

QuantBench categorizes various modeling methods from two orthogonal analytical perspectives: architectural design and training objective. The initial model suite comprises a diverse array of representative AI quant modeling methods, and we will continuously expand our repository with state-of-the-art models to ensure QuantBench remains a cutting-edge benchmarking platform. The full description of models currently supported in QuantBench can be found in the supplementary material.

## 4.1 ARCHITECTURAL DESIGN

Based on whether the model treats each asset as individual or correlated, models can be categorized as temporal and spatiotemporal. Figure 3 illustrates the evolution of these two types of models.

**Temporal Models**  Temporal models leverage the historical data of the individual assets for prediction. Representative examples include: gradient boosted tree models (e.g., XGBoost (Chen & Guestrin, 2016), LightGBM (Ke et al., 2017), and CatBoost (Prokhorenkova et al., 2018)), chosen for their robust performance on tabular data (Grinsztajn et al., 2022); recurrent neural networks (e.g. LSTM (Hochreiter & Schmidhuber, 1997)) and adaptations such as SFM (Zhang et al., 2017), DA-RNN (Qin et al., 2017), and Hawkes-GRU (Sawhney et al., 2021b)) that excel at capturing time-series dependencies; non-recurrent neural networks (e.g. TCN (Bai et al., 2018), MLP-Mixer (Tolstikhin et al., 2021)) and Transformer-based models such as Informer (Zhou et al., 2021), Autoformer (Wu et al., 2022), FEDFormer (Zhou et al., 2022), and PatchTST (Nie et al., 2022), offering alternative mechanisms for sequential data modeling.

**Spatial Models**  QuantBench incorporates several models for spatial modeling, addressing the interconnected landscape of stocks. Simple graph models such as Graph Attention Networks (GAT) (Veličković et al., 2018) and Graph Convolutional Networks (GCNs) (Kipf & Welling, 2017) are utilized to process information across market graphs effectively. Additionally, for more complex financial networks that involve various types of relationships, heterogeneous graph models such as Relational Graph Convolutional Networks (RGCN) (Schlichtkrull et al., 2018) and Relational Stock Ranking (RSR) network (Feng et al., 2019) are implemented. To capture higher-order relationships beyond simple pairwise interactions between stocks, QuantBench also includes hypergraph models such as ESTIMATE (Huynh et al., 2022), STHCN (Sawhney et al., 2020), and STHAN (Sawhney et al., 2021a), which offer a more comprehensive analysis of the financial market's structure.

## 4.2 Training Objective

The selection of a training objective is informed by task-specific requirements and the intended outcome. For example, classification tasks predict binary outcomes, such as the direction of stock price movements, which is useful for market timing strategies. Regression tasks forecast continuous outcomes such as stock returns, which are usually used for downstream portfolio optimization in stock cross-sectional strategies. Ranking focuses on the relative order of assets rather than their precise values, enhancing the profitability of characteristic-sorted portfolio. In contrast, utility maximization directly seeks to enhance financial metrics that account for both risk and return, which is usually used for end-to-end modelling that directly generates positions.

Notably, some objectives, especially those involving complex simulations or non-standard feedback mechanisms, present unique challenges for optimization. For instance, objectives involving direct financial gain, such as price advantage, are not inherently differentiable due to the inclusion of trading simulation steps. These require alternative approaches such as reinforcement learning, where a model is refined based on a reward system derived from its trading performance.

*Remark:* While we strive to faithfully reimplement models and reproduce results, discrepancies may arise between our versions and the original works. To foster transparency and community engagement, we will open-source QuantBench's implementation. We encourage contributions, including corrections from original authors and new methods, to enrich and refine QuantBench.

## 5 Evaluation

**Task-specific Metrics**  Different tasks have different goals and thus require different evaluation metrics.

- **Signal** Model outputs used as alpha signals or stock scores can guide investment decisions. Common metrics for evaluating signal quality include Information Coefficient (IC), which measures correlation between the signal and future returns, and ICIR, which adjusts IC for signal volatility.
- **Portfolio** Investment strategy performance can be measured using metrics like annualized return for profitability and risk-adjusted returns like the Sharpe ratio. Other key metrics include max drawdown (maximum loss during a period) and turnover (frequency of portfolio rebalancing).
- **Execution** These metrics assess the efficiency of trade implementation. Slippage measures the difference between expected and actual execution prices, while market impact evaluates the adverse effect of trades on market prices. Other costs such as commissions and transaction fees are also considered.

**Task-agnostic Metrics** Due to the inherent nature of quantitative finance, we consider the following additional metrics:

- **Robustness** This measures the model's performance under varying market conditions and stress tests. Key metrics include return stability, drawdown consistency, and sensitivity to input perturbations, helping assess how well the model adapts to different environments.

- **Correlation** In multi-model or multi-strategy setups, correlation metrics like pairwise correlation coefficients help assess the diversification benefits. Lower correlations indicate more diversified signals or portfolios, which can improve overall risk-adjusted returns.

- **Decay** Alpha decay tracks the diminishing effectiveness of a model's signal or strategy over time. Metrics such as the half-life of IC and time-varying performance provide insights into how quickly a model's predictive power fades and when updates might be needed.

# 6 EMPIRICAL STUDY

In this section, we present our empirical findings with QuantBench and derive some important findings and therefore potential future research directions.

## 6.1 COMPARISON BETWEEN TREE MODELS AND DEEP NEURAL NETWORKS

Table 2: Performance comparison between XGBoost and LSTM on different features

| Features | Model | IC (%) | Return (%) | SR |
|---|---|---|---|---|
| Alpha101 | XGBoost | 2.31% ± 0.00% | 24.58% ± 0.08% | 0.8093 ± 0.0029 |
| | LSTM | 4.76% ± 0.13% | 24.25% ± 3.17% | 0.7741 ± 0.0984 |
| | **Diff** | -51.47% | 1.36% | 4.55% |
| Alpha158 | XGBoost | 2.53% ± 0.00% | 20.31% ± 0.00% | 0.6407 ± 0.0000 |
| | LSTM | 5.95% ± 0.50% | 23.76% ± 5.76% | 0.7561 ± 0.1750 |
| | **Diff** | -57.48% | -14.53% | -15.27% |

Given the inherent noise in financial data, feature engineering plays a crucial role in improving the signal-to-noise ratio. This experiment examines the effect of feature engineering on model performance across tree-based and deep neural network (DNN) models. Using Chinese stock market data, we employed two feature sets—Alpha101 and Alpha158—while comparing XGBoost (tree-based) and LSTM (DNN) models. The backtest used a ranking-based stock selection strategy, selecting the top 300 stocks at each cross-section without applying feature selection. The results in Table 2 indicate that DNNs generally outperform tree-based models in IC. However, tree models produced better returns and Sharpe ratios with Alpha101. When using Alpha158, DNN performance improved, though the gains were more pronounced in IC than returns. These findings suggest that tree models tend to perform better with feature sets that have strong predictive power, potentially due to reduced overfitting or differences in problem structure (Grinsztajn et al., 2022). On the other hand, DNNs excel at capturing intricate, complex patterns, making them more adept at modeling sophisticated relationships. In future research, integrating factor mining with model design could further enhance the performance of both types of models.

## 6.2 COMPARISON AMONG DIFFERENT MODELS

This experiment explores the impact of model architecture on predictive performance. Using US stock data with volume-price and fundamental features, we trained models using IC loss and conducted backtests under the same strategy used in other experiments. Additionally, Wikidata, which provides intra-stock relational information, was incorporated. Results in Table 3 show that vanilla RNN models performed well, with slight improvements seen in adapted versions like ALSTM. In contrast, certain models, such as Hypergraph Neural Networks, failed to perform adequately, likely due to mismatches between the data type and the intended use case for these models. Transformer models designed for time-series data also underperformed in stock prediction tasks, a finding consistent with other studies. Wikidata's inclusion yielded minimal improvements, possibly because the

Table 3: Comparisons of performance of different models.

| Model Type | Model | IC (%) | ICIR (%) | Return (%) | MDD (%) | SR | CR |
|---|---|---|---|---|---|---|---|
| Vanilla RNN | LSTM | 3.89 ± 0.29 | 79.56 ± 7.72 | 54.90 ± 5.53 | -8.64 ± 1.23 | 3.0175 ± 0.2395 | 6.3922 ± 0.5040 |
| | GRU | 4.12 ± 0.12 | 81.85 ± 13.56 | 61.36 ± 4.64 | -8.44 ± 2.40 | 3.4433 ± 0.3542 | 7.6200 ± 1.7613 |
| Adapted RNN | ALSTM | 4.22 ± 0.03 | 93.47 ± 3.87 | 54.13 ± 2.91 | -8.65 ± 1.28 | 3.0361 ± 0.2256 | 6.3909 ± 1.1978 |
| | SFM | 3.58 ± 0.18 | 86.57 ± 3.55 | 53.43 ± 6.08 | -9.06 ± 0.60 | 2.9749 ± 0.3702 | 5.9478 ± 1.0961 |
| | Multi-scale RNN | 3.79 ± 0.25 | 81.55 ± 13.31 | 52.75 ± 3.37 | -9.02 ± 1.71 | 2.9540 ± 0.0769 | 5.9677 ± 0.9095 |
| | D-LSTM | 2.86 ± 0.25 | 71.36 ± 7.11 | 37.23 ± 1.57 | -9.95 ± 1.07 | 2.1891 ± 0.0957 | 3.7554 ± 0.2451 |
| | Hawkes-GRU | 3.46 ± 0.62 | 89.47 ± 33.15 | 0.35 ± 10.81 | -15.28 ± 0.87 | -0.0195 ± 0.7246 | 0.0502 ± 0.7303 |
| Other Seq Model | MSTR-I | 2.52 ± 0.30 | 52.30 ± 10.97 | 33.53 ± 4.27 | -10.48 ± 1.71 | 1.9079 ± 0.1575 | 3.2627 ± 0.6748 |
| | TCN | 3.86 ± 0.16 | 92.13 ± 10.40 | 54.72 ± 4.33 | -9.49 ± 1.72 | 2.9729 ± 0.1510 | 5.8813 ± 0.9236 |
| | Mixer | 3.67 ± 0.07 | 79.43 ± 8.71 | 46.78 ± 3.48 | -8.92 ± 2.31 | 2.7001 ± 0.3210 | 5.5299 ± 1.4799 |
| | Dlinear | 2.99 ± 0.22 | 63.39 ± 9.54 | 40.09 ± 4.53 | -7.80 ± 0.50 | 2.5229 ± 0.3029 | 5.1675 ± 0.7925 |
| Transformer | Autoformer | 0.08 ± 0.13 | 2.54 ± 4.47 | -8.82 ± 3.82 | -15.68 ± 5.05 | -0.7467 ± 0.3440 | -0.5514 ± 0.0728 |
| | FEDformer | 0.13 ± 0.08 | 2.47 ± 1.41 | -3.16 ± 1.37 | -11.12 ± 0.58 | -0.2449 ± 0.1082 | -0.2812 ± 0.1117 |
| | Pyraformer | 1.14 ± 0.00 | 12.08 ± 0.25 | 7.20 ± 7.44 | -6.55 ± 1.75 | 0.6162 ± 0.6166 | 0.9819 ± 0.8742 |
| | PatchTST | 1.07 ± 0.18 | 17.91 ± 3.07 | 8.96 ± 6.10 | -12.40 ± 2.07 | 0.6079 ± 0.4118 | 0.7806 ± 0.5557 |
| Tabular | TFT | 3.99 ± 0.05 | 80.06 ± 3.05 | 54.31 ± 5.02 | -7.10 ± 0.68 | 3.2324 ± 0.1920 | 7.6561 ± 0.0300 |
| Vanilla GNN | GCN | -0.10 ± 0.04 | -6.30 ± 2.60 | -13.07 ± 1.79 | -19.50 ± 1.91 | -1.1009 ± 0.1568 | -0.6685 ± 0.0275 |
| | GAT | 3.90 ± 0.15 | 85.48 ± 3.86 | 58.07 ± 4.69 | -8.20 ± 0.61 | 3.0730 ± 0.1991 | 7.1307 ± 0.9839 |
| Relational GNN | RGCN | 3.78 ± 0.32 | 75.62 ± 9.61 | 49.58 ± 6.17 | -8.83 ± 2.17 | 2.7706 ± 0.3492 | 5.9268 ± 1.8845 |
| | HATS | 0.23 ± 0.10 | 12.97 ± 6.03 | -11.98 ± 3.76 | -19.31 ± 2.60 | -1.0024 ± 0.3119 | -0.6110 ± 0.1372 |
| | RSR | 0.12 ± 0.08 | 5.79 ± 3.70 | -12.21 ± 4.07 | -18.62 ± 4.30 | -1.0150 ± 0.3333 | -0.6459 ± 0.0643 |
| | HATR-I | 3.07 ± 0.02 | 59.75 ± 4.35 | 37.66 ± 7.83 | -11.00 ± 5.27 | 2.1913 ± 0.5738 | 4.0609 ± 2.6585 |
| Hypergraph NN | STHAN | 0.05 ± 0.13 | 3.11 ± 6.73 | -11.44 ± 2.40 | -17.73 ± 2.35 | -0.9541 ± 0.2098 | -0.6411 ± 0.0525 |
| | STHGCN | -0.01 ± 0.04 | -0.44 ± 2.98 | -9.90 ± 0.79 | -16.41 ± 0.84 | -0.8278 ± 0.0662 | -0.6031 ± 0.0243 |
| Adaptive GNN | THGNN | 4.93 ± 0.22 | 100.27 ± 2.95 | 65.04 ± 2.01 | -9.32 ± 0.37 | 3.3184 ± 0.1365 | 6.9788 ± 0.1852 |
| | DTML | 3.87 ± 0.18 | 80.53 ± 4.49 | 54.71 ± 4.71 | -7.99 ± 0.65 | 3.1993 ± 0.1832 | 6.9029 ± 1.0647 |
| | Crossformer | 4.50 ± 0.42 | 77.54 ± 13.49 | 55.03 ± 6.50 | -9.06 ± 1.23 | 3.1269 ± 0.2902 | 6.1366 ± 0.8872 |

information is already widely known and exploited by other market participants. Relational GNNs, such as RGCN, which account for differences in edge types, were more effective than homogeneous models like GCN. Adaptive graph models, which learn graph structures from data, outperformed others, suggesting that latent relationships between stocks hold valuable predictive power. The results point to future research opportunities in designing models that can better integrate temporal and cross-sectional information through unified architectural approaches, avoiding bottlenecks between stages of information processing.

## 6.3 DIFFERENT TRAINING OBJECTIVES

Table 4: Comparison across different training objectives

| Model | Objective | Return (%) | SR | IC (%) |
|---|---|---|---|---|
| LSTM | CLF | 19.31 ± 0.80 | 1.9757 ± 0.1888 | 2.77 ± 0.32 |
| | IC | 36.03 ± 3.99 | 1.8642 ± 0.1930 | 3.62 ± 0.09 |
| | MSE | 8.97 ± 10.41 | 0.4484 ± 0.5537 | 1.78 ± 1.20 |
| | Ranking | -1.11 ± 2.74 | -0.0978 ± 0.2347 | -0.07 ± 0.09 |
| RGCN | CLF | 19.53 ± 1.48 | 2.1073 ± 0.1532 | 2.66 ± 0.11 |
| | IC | 44.97 ± 3.23 | 2.1990 ± 0.1855 | 3.97 ± 0.30 |
| | MSE | 9.13 ± 13.99 | 0.5141 ± 0.9210 | 1.47 ± 1.18 |
| | Ranking | -3.75 ± 6.26 | -0.2929 ± 0.5210 | 0.05 ± 0.03 |
| DTML | CLF | 14.10 ± 1.96 | 1.5255 ± 0.0703 | 2.55 ± 0.10 |
| | IC | 38.56 ± 11.29 | 1.8796 ± 0.6417 | 3.24 ± 0.55 |
| | MSE | -1.98 ± 1.57 | -0.1616 ± 0.1272 | 0.22 ± 0.37 |
| | Ranking | -2.21 ± 3.59 | -0.1793 ± 0.2833 | -0.07 ± 0.07 |

The choice of training objective is critical in quant modeling as it directly influences the resulting model's final performance. This experiment investigated the effect of various training objectives, including classification loss (CLF), IC loss, MSE loss, and pairwise ranking loss (Ranking), alongside a combination of MSE and ranking loss with weighted coefficients. The experimental setup mirrored that of previous sections, with the results summarized in Table 4. Different training objectives excel in different performance metrics. For instance, LSTM models trained with classification loss achieved the highest Sharpe ratio, while IC loss yielded the best IC and return metrics. This underscores the importance of tailoring the loss function to the model's final objective—whether

it's return-focused or prediction-focused. Moreover, the effectiveness of training objectives varies between models. For cross-sectional models like RGCN, IC loss proved superior, while temporal models like LSTM did not benefit as much from IC loss. This suggests that future work should focus on aligning training objectives with the specific characteristics of the model architecture. Additionally, this experiment highlights the potential for research in AutoML to optimize the selection of training objectives automatically, and end-to-end learning (Liu et al., 2023; Wei et al., 2023) to better incorporate the final goal as training objective.

## 6.4 ALPHA DECAY IN QUANT MODELS

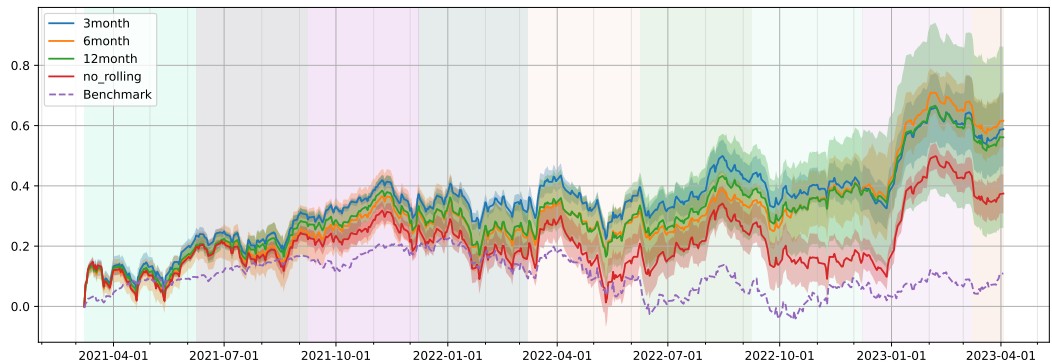

Figure 4: Comparison of different rolling schemes

In continuously evolving financial markets, alpha decay is a common phenomenon. This experiment aimed to measure the speed of market evolution by evaluating different model updating schemes using walk-forward optimization techniques with varying rolling steps (3, 6, and 12 months, as well as no rolling). The study was conducted on US stock data from 2021 to 2023, with the S&P 500 used as a benchmark. Figure 4 shows that more frequent model updates, such as the 3-month rolling scheme, consistently produced the best performance, while the no-rolling approach performed the worst. This suggests that more frequent updates help address the distribution shifts caused by market evolution, thus slowing alpha decay. However, these frequent updates come with significant computational costs, as the model must be retrained more often. This emphasizes the need for future research into more efficient online learning and continual learning techniques that can reduce the computational burden while still allowing for timely model updates.

## 6.5 HYPERPARAMETER TUNING FOR QUANT MODELS

Table 5: Comparison of different settings of validation set selection and training schema.

| Training | Segmentation | IC (%) | ICIR (%) | Return (%) | SR |
|----------|--------------|--------|----------|------------|-----|
| Normal | Tail | $3.29 \pm 0.77$ | $75.39 \pm 31.09$ | $29.87 \pm 13.00$ | $1.5454 \pm 0.7088$ |
| | Random | $3.86 \pm 0.26$ | $93.96 \pm 9.25$ | $36.84 \pm 9.27$ | $1.8969 \pm 0.4680$ |
| | Fragmented | $1.71 \pm 0.56$ | $41.58 \pm 17.52$ | $26.55 \pm 8.12$ | $1.2500 \pm 0.4283$ |
| Retrain | Tail | $3.15 \pm 0.22$ | $77.18 \pm 3.76$ | $26.62 \pm 6.06$ | $1.4431 \pm 0.2227$ |
| | Random | $3.80 \pm 0.39$ | $88.34 \pm 13.73$ | $41.36 \pm 11.08$ | $1.9334 \pm 0.4886$ |
| | Fragmented | $2.59 \pm 0.98$ | $57.05 \pm 20.24$ | $30.26 \pm 9.47$ | $1.5372 \pm 0.4348$ |

A key challenge in quantitative finance is the selection of a validation set for hyperparameter tuning. Traditionally, the validation set is selected as the tail segment of the training data just before the test set, assuming it will best reflect out-of-sample performance. However, validation set selection is not unique (de Prado, 2018), and different methods can yield varied results. In this experiment, we compared different validation set construction methods: tail, random, and fragmented (scattered fragments across the historical period). We also tested two training strategies: normal, where the validation data is not used in training, and retrain, where the model is retrained using the entire dataset (including the validation set) after hyperparameters are tuned. The results in Table 5 show

that retraining on the full dataset had little or negative impact in the tail and random settings but produced positive results in the fragmented design, suggesting that hyperparameters tuned this way may be more stable. The random validation set method outperformed the tail method, likely because it introduced more diverse data patterns into the validation process. While the fragmented setting showed potential, further research is needed to refine this approach and optimize hyperparameter stability.

## 6.6 QUANT MODEL ENSEMBLE

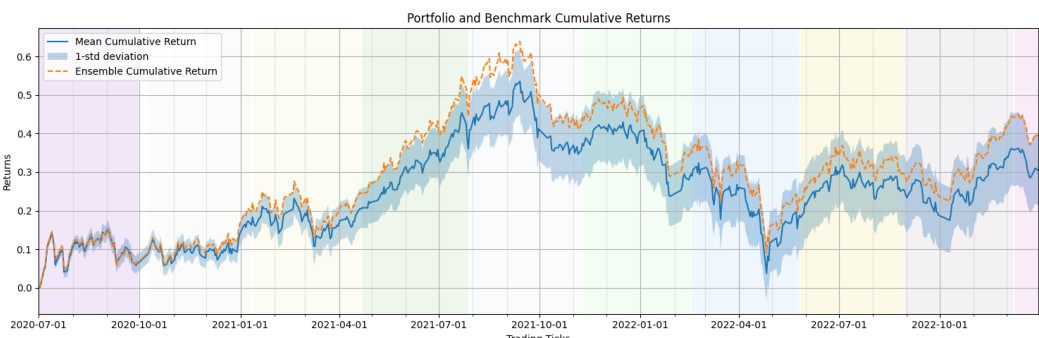

Figure 5: Ensemble curve with variance illustrated. Shaded area indicates different rolling periods.

Financial data typically exhibit a low signal-to-noise ratio, making models prone to overfitting. This experiment assessed how overfitting affects model performance and explored ensembling as a mitigation strategy. We used volume-price data from US stocks and trained an MLP-Mixer model over 40 repeated runs, each with a different random seed. An ensemble of the 40 models' predictions was computed and backtested. Figure 5 shows significant variance in performance across different runs, confirming the susceptibility of models to overfitting on noisy patterns. However, ensembling the predictions helped reduce this variance, improving robustness and protecting against overfitting. Even a simple model averaging approach provided a notable performance boost. Future research should focus on methods that encourage diversity during model training, building more robust ensembles and further mitigating the effects of overfitting.

## 7 RELATED WORKS

Quantitative modeling traditionally starts with multi-factor models. While being explainable, these conventional models relying on linear regression and predefined factors often miss capturing complex non-linear interactions among factors. In contrast, recent advancements have integrated AI into quantitative finance (Kelly & Xiu, 2023; Cheng et al., 2020; Hu et al., 2021; Sonkiya et al., 2022), utilizing techniques from gradient-boosted trees (Chen & Guestrin, 2016; Ke et al., 2017) to deep learning, which excel in identifying intricate patterns that enhance predictive accuracy. AI models (Zhang et al., 2017; Qin et al., 2017; Du et al., 2021; Wang et al., 2021a; Sawhney et al., 2021b; Deng et al., 2019; Ding et al., 2021; Feng et al., 2019; Chen et al., 2018; Wang et al., 2021b; Xu et al., 2021) are trained to forecast market trends and guide portfolio optimization through mechanisms like reinforcement learning (Jiang et al., 2017; Zhang et al., 2022; Wang et al., 2021c; 2019b) and imitation learning (Niu et al., 2022; Liu et al., 2020), reflecting a holistic approach to modern quant modeling. Moreover, the specific properties of financial data calls for relevant studies in causal inference (Zhu et al., 2021), transfer learning (Li et al., 2022), ensemble learning (Sun et al., 2023b), continual learning (Zhao et al., 2023), etc. Existing works have also explored evaluating these AI-driven methods. Qlib (Yang et al., 2020) constructs a benchmark for stock prediction, but mostly focus on temporal models and has an earlier cutoff (early 2022). FinRL-Meta (Liu et al., 2022) and TradeMaster (Sun et al., 2023a) are two comprehensive platforms for quantitative investment with a special focus on reinforcement learning. Compared with these works, QuantBench is designed to provide a holistic view of the full quant research pipeline, rather than focusing on specific techniques such as reinforcement learning.

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

## A   DISCUSSION ON LIMITATIONS AND FUTURE WORK

Despite the significant strides made with QuantBench, opportunities for enhancement remain. In terms of data, there is a need to broaden the scope by incorporating alternative data sources and to deepen the granularity by including metrics such as order flow, which would enrich the dataset with more detailed information. On the model front, integrating newer architectures and innovative formulations will further enhance the platform's capability. For evaluation, implementing more realistic and efficient backtesting methods, along with metrics tailored to specific tasks, will improve the robustness and applicability of the benchmarks.

## B   COMPARISON WITH SOTA BENCHMARKS

Table 6: Comparison with other benchmark works

| Benchmark | # Dataset | # Models | # Metrics | Data Coverage | | | | Tasks | | | |
|---|---|---|---|---|---|---|---|---|---|---|---|
| | | | | Volume-price | Fundamental | Relational | News | FM | Pred | PO | OE |
| Qlib | 2 | 18 | 7 | ✓ | ✓ | | | | ✓ | | ✓ |
| FinRL | 2 | 8 | 5 | ✓ | ✓ | | ✓ | | | ✓ | ✓ |
| TradeMaster | 13 | 16 | 10 | ✓ | | | | | | ✓ | ✓ |
| QuantBench | **24** | **44** | **20** | ✓ | ✓ | ✓ | ✓ | ✓ | ✓ | ✓ | ✓ |

## C   EXTENDED DETAILS ON DATASETS

Table 7: Coverage of different types of data in current version of QuantBench.

| | | | Volume-price | | | Fundamental | | | |
|---|---|---|---|---|---|---|---|---|---|
| | | | Daily | Minute | Tick | PIT | Industry | Relational | News |
| Stocks | US | All | ✓ | ✓ | ✓ | ✓ | ✓ | ✓ | ✓ |
| | | SP500 | ✓ | ✓ | ✓ | ✓ | ✓ | ✓ | ✓ |
| | | SP600 | ✓ | ✓ | ✓ | ✓ | ✓ | ✓ | ✓ |
| | | SP400 | ✓ | ✓ | ✓ | ✓ | ✓ | ✓ | ✓ |
| | CN | All | ✓ | ✓ | | ✓ | ✓ | ✓ | |
| | | CSI300 | ✓ | ✓ | | ✓ | ✓ | ✓ | |
| | | CSI500 | ✓ | ✓ | | ✓ | ✓ | ✓ | |
| | | CSI1000 | ✓ | ✓ | | ✓ | ✓ | ✓ | |
| | HK | All | ✓ | | | | ✓ | ✓ | |
| | | HSLI | ✓ | | | | ✓ | ✓ | |
| | | HSMI | ✓ | | | | ✓ | ✓ | |
| | | HSSI | ✓ | | | | ✓ | ✓ | |
| | UK | All | ✓ | ✓ | | ✓ | ✓ | ✓ | |
| | | FTSE 100 | ✓ | ✓ | | ✓ | ✓ | ✓ | |
| | | FTSE 250 | ✓ | ✓ | | ✓ | ✓ | ✓ | |
| | | FTSE SmallCap | ✓ | ✓ | | ✓ | ✓ | ✓ | |
| | JP | All | ✓ | | | | ✓ | ✓ | |
| | | NIKKEI225 | ✓ | | | | ✓ | ✓ | |
| | | TOPIX | ✓ | | | | ✓ | ✓ | |
| | FR | All | ✓ | | | | ✓ | ✓ | |
| | | CAC40 | ✓ | | | | ✓ | ✓ | |
| Forex | | | ✓ | | | | | | ✓ |
| Crypto | | | ✓ | | | | | | ✓ |
| Futures | | | ✓ | | | | | | |

Table 8: Major regions and stock universes involved in QuantBench

| Region | | China | | US | | Hong Kong | | UK | |
|---|---|---|---|---|---|---|---|---|---|
| Attribute | | Name | Stocks | Name | Stocks | Name | Stocks | Name | Stocks |
| | Large-cap | CSI 300 | 300 | S&P 500 | 500 | HSLI | 30 | FTSE 100 | 100 |
| Universe | Mid-cap | CSI 500 | 500 | S&P 400 | 400 | HSMI | 50 | FTSE 250 | 250 |
| | Small-cap | CSI 1000 | 1000 | S&P 600 | 600 | HSSI | 150 | FTSE SmallCap | 300 |

Table 9: Statistics of relational data on stocks. The statistics are taken with respect to the full stock universe on the regional market.

| | #Stocks | Wikidata | | | | Industry | |
|---|---|---|---|---|---|---|---|
| | | Nodes | Ratio | 1-hop edges | 2-hop edges | Industry number | Average degree |
| CN | 5128 | 728 | 14.20% | 23 | 4809 | 30 | 159.87 |
| US | 6375 | 1775 | 27.84% | 219 | 41401 | 10 | 585.63 |
| UK | 2556 | 215 | 8.41% | 10 | 9805 | 10 | 207.08 |
| HK | 2688 | 1367 | 50.86% | 98 | 6995 | 10 | 217.5 |

# D SUPPLEMENTARY EXPERIMENTAL RESULTS

## D.1 COMPARISON ON DIFFERENT DATASETS

Table 10 shows the performance of different models across four large-cap stock universes. It can be seen models show higher variances in returns and Sharpe ratio on CSI300 than on other datasets, indicating more diversified patterns captured by different models than on others. Besides the negative return on HSI may be due to the smaller size of the universe (only 50), making the portfolio strategy deteriorating to the index itself.

## D.2 MODEL CORRELATION

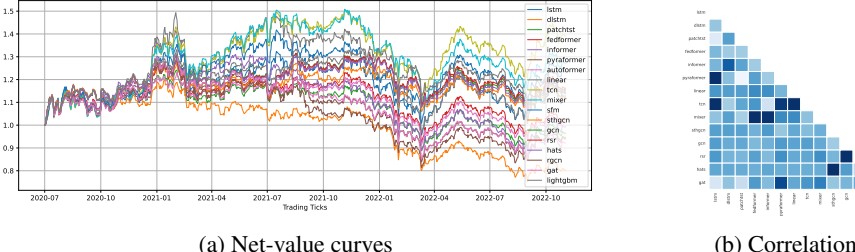

(a) Net-value curves          (b) Correlation matrix

Figure 6: Comparison of diffrent models on CSI300 dataset.

Figure 6b illustrates the correlation matrix of model predictions on CSI300. A low correlation among models can be observed and indicates potential for ensembling different model outputs. We also illustrate the variances among different repeated runs of a single MLP-Mixer model in Figure 5. The superior performance of the ensemble indicates that even for the same model, it may capture different views of the same dataset that all contributes to better predictions.

## D.3 THE EFFECT OF ADDING MORE INFORMATION

Incorporating diverse information sources is increasingly prevalent in quantitative finance. This experiment evaluates whether adding more information enhances predictive accuracy. We used five information sources: volume-price (VP), fundamental (F), news (N), industry (I), and Wikidata (W), applying two modeling techniques: XGBoost (tree-based) and a DNN. In XGBoost, all sources were used as features, while in the DNN, industry and Wikidata were represented as graphs. The data used was from the US stock market. As seen in Table 11 and Table 12, raw volume-price data had weak predictive power. Adding fundamental, news, and industry data improved performance for both

Table 10: Main results. Red represents negative value.

| Method | SP 500 | | | | CSI 300 | | | | HSI | | | | FTSE100 | | | |
|---|---|---|---|---|---|---|---|---|---|---|---|---|---|---|---|---|
| | IC | ICIR | AR | SR | IC | ICIR | AR | SR | IC | ICIR | AR | SR | IC | ICIR | AR | SR |
| LSTM | 0.12 | 1.20 | 0.17 | 0.61 | 0.05 | 1.28 | (0.07) | (0.55) | 0.10 | 1.33 | (0.02) | (0.21) | 0.08 | 1.35 | 0.33 | 1.94 |
| SFM | 0.12 | 1.21 | 0.18 | 0.68 | 0.11 | 1.28 | 0.12 | 0.39 | 0.14 | 1.30 | (0.00) | (0.16) | 0.12 | 1.28 | 0.31 | 1.69 |
| Mixer | 0.11 | 1.23 | 0.17 | 0.70 | 0.12 | 1.30 | 0.13 | 0.46 | 0.12 | 1.18 | (0.00) | (0.15) | 0.12 | 1.25 | 0.30 | 1.67 |
| TCN | 0.12 | 1.22 | 0.18 | 0.70 | 0.13 | 1.34 | 0.18 | 0.62 | 0.16 | 1.30 | (0.01) | (0.18) | 0.12 | 1.27 | 0.30 | 1.72 |
| Linear | 0.13 | 1.26 | 0.16 | 0.61 | 0.14 | 1.37 | 0.08 | 0.20 | 0.13 | 1.21 | (0.01) | (0.16) | 0.13 | 1.30 | 0.26 | 1.49 |
| Pyraformer | 0.09 | 1.17 | 0.14 | 0.60 | 0.11 | 1.25 | (0.00) | (0.17) | 0.17 | 1.32 | (0.02) | (0.23) | 0.10 | 1.27 | 0.33 | 1.89 |
| PatchTST | 0.09 | 1.14 | 0.13 | 0.50 | 0.07 | 1.24 | 0.00 | (0.15) | 0.11 | 1.25 | (0.01) | (0.18) | 0.10 | 1.30 | 0.33 | 1.87 |
| GAT | 0.16 | 1.31 | 0.16 | 0.62 | 0.15 | 1.37 | (0.01) | (0.20) | 0.14 | 1.26 | (0.01) | (0.16) | 0.16 | 1.32 | 0.32 | 1.70 |
| GCN | 0.05 | 1.36 | 0.17 | 0.73 | 0.06 | 1.40 | 0.07 | 0.21 | 0.12 | 1.46 | (0.01) | (0.18) | 0.07 | 1.26 | 0.28 | 1.62 |
| RGCN | 0.13 | 1.14 | 0.18 | 0.69 | 0.16 | 1.39 | (0.00) | (0.16) | 0.13 | 1.23 | (0.01) | (0.19) | 0.15 | 1.32 | 0.30 | 1.65 |
| HATS | 0.04 | 1.27 | 0.19 | 0.87 | 0.04 | 1.24 | 0.08 | 0.25 | 0.08 | 1.26 | (0.01) | (0.19) | 0.06 | 1.16 | 0.26 | 1.50 |
| STHGCN | 0.05 | 1.36 | 0.17 | 0.73 | 0.06 | 1.40 | 0.07 | 0.21 | 0.12 | 1.46 | (0.01) | (0.18) | 0.07 | 1.26 | 0.28 | 1.62 |

Table 11: Performance of deep neural network model on different combinations of information sources

| Info | DNN | | | |
|---|---|---|---|---|
| | IC (%) | ICIR (%) | Ret (%) | SR |
| VP | 3.43 ± 0.09 | 77.05 ± 5.73 | -0.29 ± 3.61 | -0.0193 ± 0.1961 |
| VPF | 3.56 ± 0.23 | 85.77 ± 7.44 | 30.89 ± 4.04 | 1.5592 ± 0.2115 |
| VPFN | 4.07 ± 0.21 | 90.86 ± 7.07 | 31.97 ± 10.74 | 1.6825 ± 0.4451 |
| VPFNI | 1.97 ± 1.32 | 46.85 ± 23.49 | 13.51 ± 5.97 | 0.9173 ± 0.4189 |
| VPFNW | 3.80 ± 0.18 | 76.58 ± 5.27 | 18.06 ± 17.81 | 0.9607 ± 0.9431 |

Table 12: Performance of XGBoost model on different combinations of information sources

| Info | Tree Model | | | |
|---|---|---|---|---|
| | IC (%) | ICIR (%) | Ret (%) | SR |
| VP | 0.78 ± 0.00 | 18.69 ± 0.00 | -5.87 ± 0.87 | -0.2939 ± 0.0438 |
| VPF | 1.16 ± 0.00 | 28.21 ± 0.00 | -4.65 ± 0.97 | -0.2213 ± 0.0468 |
| VPFN | 1.10 ± 0.00 | 26.35 ± 0.00 | -6.19 ± 2.51 | -0.2901 ± 0.1188 |
| VPFNI | 1.07 ± 0.00 | 28.29 ± 0.00 | 1.74 ± 0.11 | 0.0833 ± 0.0048 |
| VPFNW | | | - | |

models, though DNN performance suffered when industry data was structured as graphs due to the limitations of RGCN. Wikidata improved IC performance but reduced returns, suggesting GNNs may be more suited to portfolio optimization than direct return prediction. In summary, while more information generally improves model performance, the method of integration (features vs. graphs) is crucial. Future work should focus on optimizing models to better leverage diverse data sources.

