# OpenReview forum: "QuantBench: Benchmarking AI Modeling for Quantitative Investment"
_ICLR.cc/2025/Conference — Submitted to ICLR 2025_

### Official Review · Reviewer_2KLY · 2024-10-30

**Soundness:** 2
**Presentation:** 2
**Contribution:** 2
**Rating:** 3
**Confidence:** 4

**Summary:**

In this paper, the authors propose the QuantBench, which is a platform claimed to boost quant research efficiency by eliminating burdensome preprocessing tasks, enabling researchers to focus on algorithmic innovations. They state that it serves as a bridge between academia and industry, facilitating the implementation of advanced algorithms in investment strategies and enhancing practical applications of academic research. Through standardization, they believe that QuantBench can improve communication between these sectors, accelerating AI advancements in quantitative investment. They also conducted multiple empirical studies using QuantBench to explore key research directions, including addressing quant data's distribution shifts, evaluating graph structures, and assessing the real-world applicability of neural networks versus tree models. The outcomes of these studies highlight the needs for  robust modeling approaches, such as training models with inherent diversity or implementing causal learning methods.

**Strengths:**

The authors effectively convey the importance and necessity of having a benchmark platform like QuantBench in the realm of quantitative finance.

The abstract and introduction are well-structured and clearly articulate the research goals and the authors' approach, providing a solid foundation for understanding the study's scope and significance.

**Weaknesses:**

(1) The benchmark incorporates models that do not include some state-of-the-art (SOTA) approaches. Although the authors present a clear roadmap for model development in Figure 3, the experiments only include a limited selection of earlier models. Plus, there are more recent related work which can be included in the model comparison, like [1]. Could you please clarify why the SOTA methods are excluded from your model comparisons? The absence of these comparisons leaves the derived insights less compelling.

[1] Wang, S., Yuan, H., Zhou, L., Ni, L. M., Shum, H. Y., & Guo, J. (2023). Alpha-gpt: Human-ai interactive alpha mining for quantitative investment. arXiv preprint arXiv:2308.00016.

(2) While Section 3 of the paper extensively describes the data collection process for the benchmark, the specifics regarding the data inputs for each model type outlined in Section 6.2 remain ambiguous and potentially inconsistent. It is noted that the authors claim their selected models utilize textual feature inputs; could you provide more detailed explanations of how models categorized under 'Tabular' incorporate these textual features? Additionally, the paper mentions that graph models use graphical features as partial inputs, yet it lacks essential details. It is crucial to include information on whether other comparative models also utilize graphical features, how they do so, and where one can find details about the graph construction to assess the quality of your graph generation.

(3) And model categorization in Table 3 lacks strong reference. Would you please provide the corresponding academic reference for your model classification?

(4) Section 6.4 appears to lack crucial details, as the authors do not provide the necessary context for alpha decay, such as its typical usage and the commonly significant levels in the finance domain. Additionally, although multiple models are discussed in the previous section, it is unclear which model is used to create the visualization in Section 6.4. The authors have not clearly specified this. It would be beneficial for the clarity and completeness of the analysis if these points were addressed in the text.


(5) The shadow color choices used to represent 1 standard deviation for the two types of returns could be improved by using different colors. Currently, it is challenging for readers to distinguish between them. Using distinct colors would enhance clarity and make it easier to differentiate the returns visually.

(6) Given that the paper presents multiple studies to claim the applied scenarios of QuantBench from various perspectives, the derived insights are dispersed and challenging to follow. It is essential to include a conclusion section that consolidates these insights into a summarizing paragraph. This section should discuss how QuantBench can enhance real-world investment strategies and identify the unresolved questions that persist. Such a conclusion will provide a comprehensive overview, highlighting the platform’s practical implications in the industry and directing future research towards addressing the remaining challenges.

**Questions:**

Please refer to the question listed in the weakness section.

**Details Of Ethics Concerns:**

None.

---

### Official Review · Reviewer_LEfb · 2024-11-02

**Soundness:** 2
**Presentation:** 2
**Contribution:** 2
**Rating:** 3
**Confidence:** 2

**Summary:**

The paper provides a benchmark platform for AI methods in quantitative investment. It offers as strength "standardization with quantitative industry practices", which is not a strength in general. It might be an advantage for transferability of results, but in general industry standards can restrict innovation and are often rather complex. Another strength argued for is the integration of various AI algorithm, which should be considered a minimum requirement rather than a key strength. The last strength, i.e., full-pipeline coverage is more interesting, as other benchmarks often only provide data, putting the burden on pre-processing on the researcher, which also makes reproducibility more challenging. The paper focuses on non-GenAI (though it does assess transformers), which is ok, though it is certainly a restriction, as from personal experience I know a number of investors that use, e.g., ChatGPT for the better or worse. That is, the approach of feature engineering is very heavily embedded in the platform (e.g., they provided extracted features from news), though overall receiving less and less attention in AI. Also the data is only vaguely described in Section 2 and 3, leaving the reader puzzled about the basic composition of the benchmark.

A major concern is that the paper fails to discuss any ethical (and regulatory) concerns, such as bias (or compliance, which is huge topic in industry, which the seem to aim at).

**Strengths:**

see above

**Weaknesses:**

see above

**Questions:**

None

---

### Official Review · Reviewer_ccwr · 2024-11-04

**Soundness:** 3
**Presentation:** 3
**Contribution:** 3
**Rating:** 6
**Confidence:** 4

**Summary:**

The authors introduce a new industrial-grade benchmark with three goals in mind: (a) alignment with quantitative investment industry practices, (b) flexibility to integrate artificial intelligence algorithms, and (c) provide a pipeline that covers the whole quantitative investment process. They also perform an empirical study that  reveal some new research directions.

**Strengths:**

- the authors propose a novel benchmark for artificial intelligence methods on quantitative investment
- they compare the proposed benchmark with existing SOTA benchmarks
- they perform a comprehensive evaluation of machine learning methods related to quantitative investment

**Weaknesses:**

- the authors make some claims that are not backed by substantial experimentation
- the authors should include some assessment regarding the statistical significance of the results obtained

**Questions:**

Dear authors,

We consider the work interesting and relevant. Nevertheless, we would like to point out certain improvement opportunities.

GENERAL COMMENTS

(1) - "The results in Table 2 indicate that DNNs generally outperform tree-based models in IC" -> The authors only compare an LSTM model against an XGBoost model. We consider the claim to be too strong for the referenced results. To strengthen the claim, we encourage the authors to conduct experiments with additional tree models (e.g., random forest, LightGBM, CatBoost, ...) and contrast them to the results for deep learning models mentioned in Table 3.

(2) - We encourage the authors to include some statistical tests to assess whether the differences among results are statistically significant. This could be an important step in the benchmark when analyzing the results.

(3) - The authors mention three task-agnostic metrics (robustness, correlation, decay) but it is unclear where they were reported and assessed when considering the results obtained.

(4) - The authors provide a timeline of models covered by QuantBench in Figure 3. Nevertheless, the experiments do not cover some of the most recent ones. Would it be possible to include them?

TABLES

(5) - All tables reporting metrics: (a) provide arrows next to the metric names (up/down) indicating whether a higher/lower result is better; (b) bold the best results, (c) align numbers to the right, to make differences of magnitude evident to the reader.

MINOR COMMENTS

(6) - "charaterstic-sorted" -> "charateristic-sorted"

(7) - "Wikidata’s inclusion yielded minimal improvements, possibly because the information is already widely known and exploited by other market participants." -> Please provide some additional context to understand how the fact that other market participants could exploit such information affects the results or the data used to train the model.

(8) - "QuantBench is designed to provide a holistic view of the full quant research pipeline, rather than focusing on specific techniques such as reinforcement learning" -> It seems that the current version of the proposed QuantBenchmark focuses only on a specific type of methods (e.g., no reinforcement learning methods were considered). We encourage the authors to provide a more detailed perspective on what is currently supported and the ambition to support in the future. Are there some kinds of models that will not be considered in the future (e.g., reinforcement learning)?

---

### Official Review · Reviewer_wmFd · 2024-11-04

**Soundness:** 3
**Presentation:** 3
**Contribution:** 2
**Rating:** 3
**Confidence:** 4

**Summary:**

This paper presents QuantBench, an industrial-grade benchmark platform designed to standardize and enhance AI research in quantitative finance. The platform integrates diverse data sources, including market, fundamental, relational, and news data, and providing a layered approach that spans alpha factor mining, modeling, portfolio optimization, and order execution. It also supports multiple data resolutions, from quarterly to tick-level data, facilitating a range of quant tasks and enabling multi-scale strategies.

QuantBench categorizes models based on architectural design and training objectives, evaluating temporal models for time-series data and spatial models for relational data. It also examines the impact of training objectives, including classification, IC, MSE, and ranking losses, finding that different objectives yield varied results based on model architecture.

Experiments demonstrate that model updating frequency, validation set construction, and ensemble methods significantly impact model performance, particularly in managing alpha decay and reducing overfitting, which suggests future directions in continual learning, robust validation strategies, and ensemble diversity to further improve predictive accuracy and model stability.

QuantBench aims to provide a standardized and open-source platform that fosters collaboration and bridges the gap between academic research and practical industry applications in AI-driven quantitative investment.

**Strengths:**

1. Originality: The paper tried to provide a standardized, open-source environment specifically designed for quantitative finance, which supports quant trading tasks from factor mining to order execution, representing an integration of existing modeling methods with advanced data processing and evaluation mechanisms.

2. Quality: The quality of the work is OK, providing comparison across multiple dimensions, including model architecture, training objectives, and validation strategies.

3. Clarity: The paper is well-organized and explains the design and objectives of QuantBench.

4. Significance: The platform currently focuses on established techniques and common performance metrics, providing a useful standardized framework for evaluating models in quantitative finance, and its significance might be impactful within the industry.

**Weaknesses:**

1. The paper's originality may be somewhat limited in terms of introducing novel methodologies or fundamentally new problem formulations. Although this application to quant finance is useful, the overall concept of a benchmarking platform is not entirely new. Moreover, the paper is limited by its focus on classical machine learning and deep learning algorithms, without including more recent advancements in AI specifically tailored to quantitative finance, such as Large Language Models (LLMs) and Deep Reinforcement Learning (DRL).

2. The paper’s focus on predictive accuracy and return metrics is valuable, but it lacks depth in evaluating risk management, a crucial aspect of quantitative trading. Since QuantBench aims to align with industry standards, it would benefit from incorporating a broader set of risk metrics beyond Sharpe ratios—such as maximum drawdowns, downside risk, or conditional value-at-risk (CVaR)—to provide a more comprehensive assessment of model robustness.

3. The platform currently focuses on established techniques and common performance metrics, may not fully address the rapidly evolving needs of practitioners and researchers looking for innovative solutions in quantitative trading.

**Questions:**

1. Given the increasing use of Large Language Models (LLMs) and Deep Reinforcement Learning (DRL) in quantitative finance, does the team have plans to integrate these techniques into QuantBench in future versions?

2.  Would the authors consider expanding the platform to include risk-evaluation metrics? Could the author provide insights into the technical feasibility of incorporating stress testing in QuantBench, perhaps by simulating market shocks based on historical crisis periods?

---

### Meta-Review · Area_Chair_aNcd · 2024-12-10

**Metareview:**

This paper proposes a new quantitative trading benchmark, which consists of numerous trading-specific datasets. The paper then applies several classification / regression techniques over the data, and present experimental results, and attempts to provide analyze these results.

## Strengths
* This is one of the first ML conference papers to propose a quantiative trading benchmark, and can be considered novel due to proposing a new application to study.

## Core Weaknesses
* The paper mainly "gives a bunch of numbers" (from applying various models on financial data) without convincing takeaways or analysis - I personally haven't learned anything from the experiments. Examples of boilerplate conclusions:
  * L362: "...findings suggest that tree models tend to perform better with feature sets that have strong predictive power...On the other
hand, DNNs excel at capturing intricate, complex patterns, making them more adept at modeling
sophisticated relationships."
  * L374: "vanilla RNN models performed well, with slight improvements seen in adapted versions like ALSTM...models such as Hypergraph Neural Networks, failed to perform adequately, likely due to mismatches between the data type and the intended use case for these models"
* The paper's models aren't up-to-date with GenAI techniques, but rather are mostly classification / regression models. According to reviewers, this makes the paper lag behind the current SOTA methods in finance (whatever they may be).

Due to the core weaknesses, this is a clear reject.

**Additional Comments On Reviewer Discussion:**

The three reviewers who gave the rejection score of 3 give common criticisms:
* This paper doesn't seem to be too useful for quantiative traders - all of the results have been "a bunch of models applied to a bunch of data" without any new insights. More quant-specific subjects such as "risk management" are not discussed, which presumably would be more valuable.
* The paper doesn't include any up-to-date methods, and especially doesn't include GenAI techniques. Thus it is unlikely this paper would be valuable to any quant practitioners.
* Some key details are lacking, e.g.
  * What is alpha decay?
  * Vague discussion of data sources / description
  * How is textual data used?

Reviewer ccwr gave an accept score of 8 primarily because this paper encourages more benchmarking on quantitative trading.

While it is commendable that this paper is one of the first machine-learning submissions to propose quant benchmarks, a reader wouldn't learn very much from this paper, due to its lack of clear analysis and innovations in methods.

---

### Decision · Program_Chairs · 2025-01-22

Reject